# Effects of Different Types of High-Intensity Interval Training (HIIT) on Endurance and Strength Parameters in Children and Adolescents

**DOI:** 10.3390/ijerph19116855

**Published:** 2022-06-03

**Authors:** Thomas Bossmann, Alexander Woll, Ingo Wagner

**Affiliations:** 1Department of Sport and Sport Science, Karlsruhe Institute of Technology (KIT), Engler-Bunte-Ring 15, 76131 Karlsruhe, Germany; ingo.wagner@kit.edu; 2Department of Sport Pedagogy, Karlsruhe Institute of Technology (KIT), Kaiserstraße 12, 76131 Karlsruhe, Germany; alexander.woll@kit.edu

**Keywords:** HIIT, circuit, physical education, health, fitness

## Abstract

High-Intensity Interval Training (HIIT) promises high training effects on aerobic fitness in children, adolescents and adults in a relatively short time. It is therefore well-established in professional training settings. HIIT methods could also be suited to Physical Education (P.E.) lessons and contribute to students’ health and fitness. Since HIIT sessions need little time and equipment, they can be efficiently implemented in P.E. However, there are few studies which have examined non-running-based HIIT programs in the school sport setting. We therefore conducted an intervention study including 121 students aged 11–15 attending a secondary school in Baden Württemberg, Germany. The effects of three different forms of HIIT training varying in duration and content (4 × 4 HIIT, 12 × 1 HIIT, CIRCUIT) were analyzed. The training was conducted twice a week over 6 weeks (10–12 sessions). Strength and endurance performances were determined in pre- and posttests prior to and after the intervention. Results verified that all three HIIT programs led to significant improvements in aerobic fitness (*p* < 0.001; part ŋ^2^ = 0.549) with no significant interaction between time x group. In contrast to the running-based HIIT sessions, CIRCUIT training also led to significant improvements in all of the measured strength parameters. Retrospectively, students were asked to assess their perception of the training intervention. The HIIT sessions were well-suited to students who considered themselves as “athletic”. Less athletic students found it difficult to reach the necessary intensity levels. The evaluation showed that endurance training conducted in P.E. lessons needs a variety of different contents in order to sufficiently motivate students. Students perceiving themselves as “unathletic” may need additional support to reach the required intensities of HIIT. Circuit training sessions using whole-body drills can be efficiently implemented in the P.E. setting and contribute to students’ health and fitness.

## 1. Introduction

Children and teenagers should be more physically active than they are. Worldwide, 81% of children and teenagers do not reach the WHO recommendations for moderately intensive aerobic activity and strength training according to their age group [1]. In Germany, only around 10% of the girls and 17% of the boys are sufficiently physically active [2]. Woll et al. [3] speak of an epidemic of physical inactivity, which includes a high amount of time that adolescents spend sedentarily. During the COVID-19 pandemic in 2020, the time that young people spent in organized sports settings decreased significantly, especially in the age group between 11 and 17 [4].

With increasing age, the amount of young people who meet the WHO recommendations decreases further [5]. Longitudinal studies could show a decline by 10% each year during adolescence [6].

Accordingly, the motor and conditional abilities—specifically strength and endurance capacities—of children and teenagers in Germany are underdeveloped. They deteriorated by around 10% from 1975 to 2003 [7,8], before stagnating at a relatively low level until today [9,10]. Negative impacts on physical, psychosocial and mental health are the likely consequences [11,12]. Physical activity and fitness interventions have only been moderately effective in this context [13].

Schools and physical education lessons can contribute to students’ health preservation and improve their conditional abilities. Yet, physical activity levels within P.E. lessons are generally low. Related to an entire P.E. lesson, actual exercise and movement time for individual students lies between 18 and 50%, depending on individual performance levels and the location of the lessons [14,15,16]. Accordingly, in two consecutive P.E. lessons with an official duration of 90 min, the average student is physically active for around 20–45 min.

During this activity time, the intensity and individual training load is usually low. Breithecker et al. [14] analyzed the heart rates of 251 male and female students from grades 5 to 10 during 40 soccer lessons and found an average heart rate of 146 bpm. Other empirical data suggest lower values during lessons with different activities [17]. According to Hottenrott and Gronwald [18], an intensity above the anaerobic threshold requires heart rates of above 170 bpm.

Additional factors such as few P.E. lessons per week, poor facilities, a high number of students per grade, a lack of space in sports halls and manifold curriculum requirements can undermine the goal of improving aerobic fitness levels in students. Based on the current state of research, three P.E. lessons per week could provide 30–60 min of actual exercise time for students. In order to induce training effects with two P.E. lessons per week, highly efficient and intensive training methods are required. In order to create a lasting motivation for endurance training, these training programs need to increase students’ perceived level of competence by causing desirable effects on individual performances in a relatively short time [13,19].

A growing body of literature supports the efficiency of HIIT training sessions when it comes to improving physical and psychological health-related outcomes [13,20,21]. HIIT training consists of high-intensity exercise bouts interspersed by rest periods between exercises. As a form of ‘Intensive Interval Training’, it is characterized by a high intensity (90–95% of HR max) and activity intervals lasting from 15 s up to 4 min [22,23,24].

Even though the total training volume of HIIT sessions is generally considerably lower than that of ‘High-Volume (low-intensity) Training’ without recovery breaks (<65% of HR max, blood lactate levels < 2 mmol/L, training duration > 30 min), comparative studies often proved similar or even better effects on endurance capacity and maximal oxygen uptake capacity when HIIT sessions were used. These studies were conducted with moderately trained children, teenagers and adults [23,24,25,26,27]. Additionally, highly trained endurance athletes, who predominantly use ‘High-Volume low-intensity Training’, seem to profit when at least regularly implementing HIIT protocols in their training schedule [24].

The efficiency of HIIT protocols can be explained with adaptations of the myocardium caused by volume-induced stretching of the heart muscle combined with increased resistance during heart strokes at high intensities [24]. According to Wahl et al. [24], a precondition for effective adaptations of endurance performance through HIIT is a high intensity of 90–95% of an individual’s maximum heart rate.

With regard to the health benefits of physical fitness and the detrimental effects of inactivity and sedentariness, P.E. lessons are especially challenged to help children and teenagers efficiently improve their physical abilities. HIIT training sessions can be conducted in a relatively short time compared to traditional aerobic training methods and seem to be predestined for endurance and fitness training in P.E. lessons.

However, there are few studies on HIIT training focused on pupils and P.E. lessons [23,28]. Research on the efficiency of different and non-running-based HIIT protocols in P.E. lessons is still scarce to non-existent.

The selection and choice of specific training parameters (duration and number of intervals) when using HIIT protocols seems arbitrary, and there is no gold standard related to training parameters. Several studies found optimal effects of intervals lasting between 2 and 5 min [24,26,29]. Other authors recommend shorter intervals between 15 and 30 s [30,31]. Only recently, anecdotal reports on the efficiency of ultrashort and highly intensive intervals used in professional swimming evoked controversial debates on the topic of interval duration [32]. Peer-reviewed published literature on ultrashort training loads of 5–70 s is currently lacking, and only a few older studies addressed the topic. Helgerud et al. [26] found similar training effects when using 15 s intervals compared to 4 min intervals.

The optimal amount of intervals used in training sessions depends on the training intensity and the duration of intervals. Suggestions of between 4 and 47 repetitions can be found in the sports training literature [24]. The current literature on ultrashort intervals suggests 20–50 intervals [32].

The ideal ratio between training and recovery intervals within one interval training session is usually defined as 1:1 or 2:1 [24]. When ultrashort intervals are being used, the ratio may shift towards more training and less recovery time within a training session [31].

Research on optimal training methods is usually conducted with adult athletes or adolescent competitive athletes. Yet, the particular conditions of competitive sport settings are fundamentally different from the ones found in P.E. lessons. Accordingly, data from previous studies conducted with (endurance) athletes cannot necessarily be transferred to the school setting and its students.

The presented study is aimed at closing this research gap by comparing the effects of three 6-week-long endurance training interventions (4 × 4 min of HIIT-Training, 12 × 1 min of HIIT Training, 12 × 1 min of Circuit Training using whole-body drills) on endurance and strength parameters in a regular school sport setting.

Although the efficacy of HIIT sessions is well established, the majority of HIIT studies have examined running-based programs [6]. Combining HIIT with body weight resistance exercises can have additional benefits on fitness outcomes in adolescents [6].

Assuming that the main stimulus for an increase in stroke volume and endurance performance is training intensity [24], the underlying research hypothesis is that a circuit training session using whole-body drills in ultrashort and highly intensive intervals (12 × 1 min) may reach similar training effects (endurance performance/strength parameters) to typical running-based HIIT interventions (4 × 4 min/12 × 1 min).

Research conducted with children and teenagers suggests that HIIT training interventions implemented within a time frame of 4–5 weeks result in performance improvements of 4–7% [26,33,34]. Therefore, untrained children and adolescents should be able to improve their endurance capacity by at least 5% after 10–12 training sessions performed within 6 weeks. Changes in performance underneath this benchmark will be considered as a result of day-to-day variances and therefore independent of the training stimulus.

## 2. Materials and Methods

A school-based randomized controlled trial was conducted with female and male children and adolescents attending grades 5–8 in a secondary school in Baden Württemberg, Germany. Initially, 136 students (75 boys, 61 girls) took part in the study. Due to absence during training sessions, data from 121 students (69 boys, 52 girls) were included in the final analysis. All participants were healthy, and neither musculoskeletal injuries nor neurological or cardiovascular restrictions were prevalent.

Permission to conduct this study was granted from the relevant educational organizations. In advance of the study, children and their parents had been fully informed of the goals and content of the project in written form. The study protocol complied with the ethics code of the World Medical Association for experiments with human beings, the Declaration of Helsinki [35].

All tests and training sessions were conducted during regular P.E. lessons that took place twice a week and were taught co-educationally.

The study initially started with a pretest on aerobic endurance performance and strength capacity. After the pretests, six participating classes were randomly assigned into one of the three training groups (the P.E. teachers drew lots). During the following six weeks, the students carried out the allocated training protocols (10–12 sessions) twice a week during their regular P.E. lessons. The study was terminated with a posttest including the exact same test parameters as the pretest.

After each lesson, students gave written feedback on their level of motivation and rating of perceived exhaustion during the training session. Upon completion of the study, students rated their overall level of motivation and could suggest improvements regarding the training design. Finally, students stated whether they perceived themselves as “rather sporty and athletic” or “rather unathletic”.

All pre- und posttests and each training session were conducted by P.E. teachers at the given school. The teachers had been fully informed on the content of the study and the criteria of the tests in advance of the study.

Only data from students who completed at least 10 training sessions and both aerobic performance tests were included in the statistical analysis.

Based on the suggested work–rest ratio, the ratio for both the CIRCUIT training and the two HIIT sessions was approximately 1.5:1.

### 2.1. Pre and Posttest

The pre- and posttests were conducted after a standardized warm-up (5 min of slow running and 3–5 stretching exercises). The following test procedures of the German Motoric Test [36] were applied (see Figure 1):Number of pushups carried out within 40 s.Number of sit-ups performed within 40 s.Distance covered with a Standing Long Jump.Number of lateral side jumps (two-footed) conducted within 15 s.Running performance determined via Luc Legèr’s shuttle run test [37].

According to Bös and Tittlbach [38], the DMT has the following test quality criteria: objectivity: mean value of correlation coefficient (r) for all 8 test items = 0.95; test–retest reliability: r = 0.82; content-related validity (expert rating on a scale from 1 (very good) to 5 (poor): 2.1 for validity, 1.8 for feasibility.

The shuttle run test is recommended by the Institute of Medicine (IOM) [39] as the most accurate and appropriate filed-based measure of cardiorespiratory fitness in young people and shows a high degree of standardization. The test–retest reliability is stated with a medium coefficient of r = 0.86. The correlation with the maximal oxygen uptake capacity is r = 0.73 [40].

The students had to run back and forth on a 20 m course that was marked with two tags. Acoustic beeps signaled when the tags had to be reached and defined the speed with which the students had to run. The speed increased with time and started at 8 km per hour (km/h). Every minute, the running speed was increased by 0.5 km/h. The acceleration was signaled by shortened intervals between the acoustic signals. When students were running too fast, they had to wait at the tag for the next signal to occur. The test was terminated when the tag could not be reached three consecutive times in a row.

The maximal running speed (max. speed = MS) was converted into maximal aerobic speed (MAS) according to a formula introduced by Billat and Koralsztein [41]:MAS = 2.4 × MS − 14.7

### 2.2. Training Intervention

Each training session was protocolled by the P.E. teachers and conducted after a standardized 3–5 min warm-up session. Data collection took place from October 2021 to April 2022. The average duration of the 4 × 4 HIIT sessions was 28 min, the 12 × 1 HIIT sessions and the CIRCUIT training (see Appendix A and Appendix B) lasted approximately 20 min. The students were encouraged to maintain a high intensity of about 90% of the HR max. During the breaks between the intervals, students could either walk at their own speed, rest or drink something.

There were no changes in the design of the lessons in which the training sessions were conducted. After each lesson, students estimated their motivation during the training session and rating of perceived exhaustion (RPE) on a scale from 0 to 10 using the modified BORG-CR scale [42].

### 2.3. Statistical Analyses

All data are depicted as mean values +/− standard deviation (Appendix B). The differences between pre- and posttests had been checked for normal distribution using the Shapiro–Wilk test, and the data used was checked for equality of error variance with the Levene test.

Changes in performance between pre- and posttests (MAS/Strength parameters) within and between the three training groups were analyzed via t-tests and mixed variance analysis (ANOVA) with repeated measurement for dependent samples. The level of significance was set at *p* < 0.05, and the tests were conducted one-sidedly since improvements were to be expected.

The statistical analysis was conducted with SPSS, Version 25 (IBM, Armonk, NY, USA).

## 3. Results

All participants that were included in the study accomplished at least 10 training sessions within a time slot of 6 weeks. Seven students (two from the HIIT group, two from the 12 × 1 group and one from the CIRCUIT group) had to be removed from the study due to a lack of training sessions or because of absence during performance tests. Fifteen students (five from the HIIT group, five from the 12 × 1 group and five from the CIRCUIT group) completed both endurance tests but were missing at least one of the strength tests. These students were excluded from the analysis on changes in strength performance for the given tests.

### 3.1. Shuttle Run Test

The training group HIIT (4 × 4) reached a performance increase in maximal aerobic speed (MAS) of 9.1%, the HIIT (CIRCUIT) group of 9.4% and the HIIT (12 × 1) of 12.1%. The increase in aerobic capacity through the training intervention was statistically significant for all three groups:4 × 4 HIIT group, t(41) = −7.98, *p* < 0.001;HIIT CIRCUIT group, t(40) = −6.35, *p* < 0.001;12 × 1 HIIT group, t(40) = −5.30, *p* < 0.001.

There was no statistically significant interaction between time and group, Greenhouse–Geisser F(2.0, 0.896) = 1.37, *p* = 0.259, partial ŋ^2^ = 0.026.

### 3.2. Strength Parameters

In the 4 × 4 HIIT group, significant improvements were reached only with regard to sit-ups, t(32) = −3.30, *p* < 0.001. No significant improvements in the three other strength tests could be shown.

In the 12 × 1 HIIT group, statistically significant improvements were reached in lateral side jumps, t(31) = −2.70, *p* = 0.013 and pushups, t(31) = −3.00, *p* = 0.004. No significant improvements could be shown in the two remaining strength tests.

In the CIRCUIT group, students reached significant improvements in all four strength tests:Pushups: t(35) = −2.69, *p* = 0.005;Sit-ups: t(35) = −2.46, *p* = 0.009;Standing Long Jump: t(35) = −3.00, *p* = 0.002;Lateral side jumps: t(35) = −8.30, *p* < 0.001.

There were no significant differences in the rating of perceived exhaustion and motivation between the three groups. The students reached an average score of 5.4 on the modified BORG-CR scale, rating the intensity level as “hard”. Students who considered themselves as “athletic” rated their level of exhaustion significantly higher than those students who considered themselves as “unathletic” (t(106) = −1.14, *p* = 0.048). The distribution of ”unathletic” versus ”athletic” students showed no significant differences between the three groups. Results are depicted in Figure 2, Figure 3, Figure 4, Figure 5 and Figure 6 and Table 1.

## 4. Discussion

The aim of this study was to analyze the effects of three different High-Intensity Interval Training protocols conducted during regular P.E. lessons on aerobic endurance capacity and strength parameters in students aged 11–15.

The main findings were the following: (1)In all three training groups (CIRCUIT, HIIT 4 × 4, HIIT 12 × 1), the endurance capacity of the students increased significantly through the training intervention (MAS↑ 9–12%);(2)The 4 × 4 training group reached significant improvements in one strength test (sit-ups);(3)The 12 × 1 training group reached significant improvements in two strength tests (sit-ups, pushups);(4)The CIRCUIT training group reached significant improvements in all four of the used strength tests (sit-ups, pushups, standing long jump, lateral side jumps).

Accordingly, the research hypothesis that CIRCUIT training sessions using whole-body drills in ultrashort and highly intensive intervals may reach similar training effects on aerobic fitness compared to typical running-based HIIT interventions could be confirmed.

The study was conducted in a school setting, which is characterized by a large heterogeneity of students’ fitness levels. The pretests could show that the mean aerobic endurance capacity of the groups were characteristic of an average population of children and adolescents aged 11–15 [43]. Previous studies conducted with children and adolescents outside the school setting found similar or slightly lower improvements of aerobic endurance capacity when running-based HIIT protocols had been used [23,34,44,45,46]. A systematic review on the effectiveness of running-based HIIT sessions reported volatile performance improvements ranging from 2% to 43% in young adults, depending on the initial fitness level of participants and the duration and design of the intervention [21,34,47]. Beneficial effects of HIIT sessions in comparison to volume-oriented endurance training methods seem likely, especially when longer interventions and intervals and a greater work–rest ratio are used.

A variation of training content in a moderately to highly intensive circuit training session could show similar effects on aerobic fitness with even shorter intervals. Jeneviv et al. [48] analyzed the effects of 18 circuit training sessions conducted with obese undergraduates within 6 weeks. The moderately intensive (50–70% of HR max) intervals lasted 60 s (with 20 s breaks between the intervals) and included both resistance and aerobic exercises similar to the CIRCUIT training group in this study. The authors reported an increase in VO_2_ max of around 10%. A meta-analysis on the effects of resistance training on VO_2_ max in different populations could show similar effects [49]. Costigan et al. [6] evaluated the effects of 24 HIIT sessions carried out with 65 pupils within 8 weeks. Each session lasted from eight to ten minutes, and the work to rest ratio was 30:30 s. The authors used similar exercises as the CIRCUIT group (shuttle runs, jumping jacks, skipping, squats, pushups, hovers). Cardiorespiratory fitness was assessed using the shuttle run test [37]. The aerobic capacity of the pupils increased by 6%. Improvements with regard to strength parameters (standing long jump/pushups) were marginal and non-significant.

Contradictory findings were published by Schmidt et al. [50] who investigated ninety-six recreationally active college-aged subjects and analyzed the effects of 24 seven-minute circuit training workouts conducted within eight weeks (three workouts per week). The results suggest that short-duration, high-intensity circuit training may improve muscle endurance in a moderately fit population but not give statistically significant improvements in aerobic capacity.

Various peripheral and central physiological factors may explain enhanced VO_2_ max levels after HIIT interventions. Besides increased blood and stroke volume through adaptations of the myocardium [24,26,51], the combination of high lactate values and hypoxia may lead to the regeneration of mitochondria and peripheral vessels and thereby improved capillarization [52]. Some authors assume a more efficient glycogen supercompensation through HIIT in comparison with less intensive volume-oriented training sessions [24]. The suggested peripheral adaptations may explain performance improvements in the strength endurance tests in the two running-based HIIT groups. However, in the tests that required explosive strength capacity, only the CIRCUIT training group showed performance improvements.

Cardiological adaptations through HIIT sessions may explain why intensive whole-body drill exercises can lead to similar adaptations of aerobic fitness compared to running-based programs. Training intervals can therefore be varied and adapted in a multi-faceted way to the school sport setting, as long as a high enough intensity can be reached.

### 4.1. Practical Consequences for the School Sport Setting

The perceived motivation of the participating students during the training sessions was moderate (mean value: 2.1 on a Likert scale ranging from 0 to 3). Students perceiving themselves as “sporty” and “athletic” were significantly more motivated during the lessons than students who considered themselves as “unathletic”. This difference was prevalent in all three training groups. Additionally, more athletic students reached a significantly higher intensity level during the HIIT sessions in all three groups. Some authors argue that inactive people and people leading a sedentary lifestyle might experience feelings of incompetence and failure during highly intensive training sessions, resulting in demotivation and even lower physical activity levels [53]. Previous studies have acknowledged the experience of enjoyment and motivation as critical for motivating students to continuously participate in activity environments [54]. It therefore appears crucial for the school sport setting to provide training conditions that allow students to successfully increase their perceived level of competence and thereby meet their basic psychological needs during the lessons [54]. It also seems essential to choose training settings in which the difficulty of the chosen contents and the level of intensity are manageable, adaptable and motivating for all students of a class.

The evaluation of the different training programs revealed that students perceived the variation of training content and the use of music as motivating and helpful in order to sustain high intensity levels. Shorter intervals seemed to be marginally more motivating for the students.

Implementing challenging obstacle courses or circuit sessions in which students could individually adapt or choose exercises might increase the level of motivation and especially help “unathletic” students who find it hard to reach high intensity levels.

The presented study suggests that HIIT sessions including highly intensive whole-body drills lead to significant adaptations in endurance and strength capacity and show similar results to running-based training programs. With regard to the manifold demands that school curricula lay on P.E. lessons and additional challenges teachers face in the school sport setting, it seems evident that P.E. lessons require efficient, time-saving training methods such as HIIT. Arrangements such as circuit training sessions seem particularly suited to simultaneously combining different training and fitness goals.

### 4.2. Limitations

The presented field study was conducted during regular P.E. lessons. Accordingly, it was impossible to randomly assign each individual student according to his or her initially documented aerobic fitness level. Slight differences in performance capacity between the three groups had to be accepted, as in previous studies in the school setting [21,23]. Secondly, P.E. groups are highly heterogeneous. Even though students had been advised and encouraged to execute the training sessions with an intensity near their individual limit, especially “unathletic” students were unable to reach the required intensity levels. If high intensities could not be reached by a student, he or she continued training at the highest achievable speed. The training protocols used confirmed an altogether high intensity on the modified BORG scale. Yet, it is outright impossible to ensure heart rate values above 90% of the individuals’ maximal heart rates during P.E. lessons. In more competitive sport settings, this precondition is more likely achievable. Reliable heart rate monitoring during tests and training sessions could ensure a more objective recording of training intensity and the level of exhaustion the participants reach.

Additionally, complex diagnostic procedures which may assess students’ fitness and health related parameters more accurately cannot be conducted in the school setting due to ethical reasons. However, indirect performance measurements such as Léger’s Shuttle Run Test reliably indicate an increase in aerobic capacity. The test was specifically developed for children and teenagers and shows a high accuracy when predicting VO_2_ max [55].

Future studies may add a control group in their study design. Students are used to get marked and evaluated regularly. Especially repeated tests in the P.E. setting may evoke the assumption among students that performance improvements could be evaluated or expected, so that their willingness to strain themselves to the fullest might have been increased in the second testing procedure. Learning effects may further have increased this bias [6].

The present study shows that HIIT sessions efficiently improve aerobic fitness in students, even though training intensity could not perfectly be controlled. CIRCUIT training has reached similar effects to the running-based programs and has added additional improvements in strength related performances. Since high endurance capacity is associated with various adverse health outcomes and reduced mortality [56], P.E. lessons may significantly contribute to students’ health if HIIT sessions are being implemented.

## 5. Conclusions

Fitness training carried out during P.E. lessons faces multiple challenges with regard to time and resources. Complex curricula demands may further undermine the goal of efficiently increasing VO_2_ max and fitness in students.

The preconditions of the school setting as well as current research data seem to indicate that HIIT protocols seem better suited to P.E. lessons than volume-oriented endurance training methods, which require at least 30% more time. Evidence from this study highlights the potential of implementing circuit training sessions within P.E. to improve fitness outcomes.

The long-term effectiveness of the chosen training sessions should be evaluated in future studies with longer follow-up periods. Additionally, longer HIIT sessions that solely use running-based intervals should be examined with regard to possible negative consequences such as overreaching or injuries.

## Figures and Tables

**Figure 1 ijerph-19-06855-f001:**
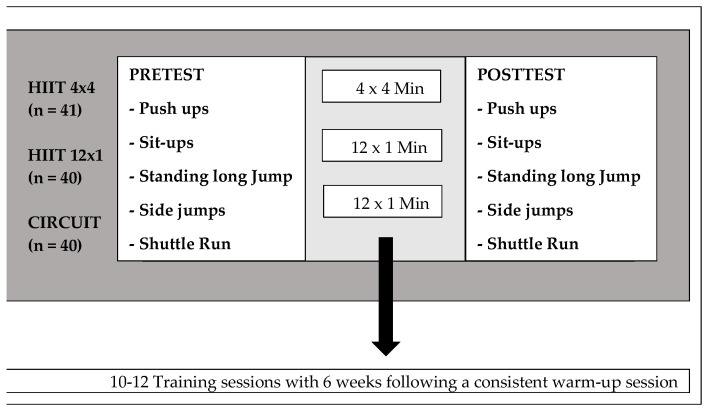
Study design.

**Figure 2 ijerph-19-06855-f002:**
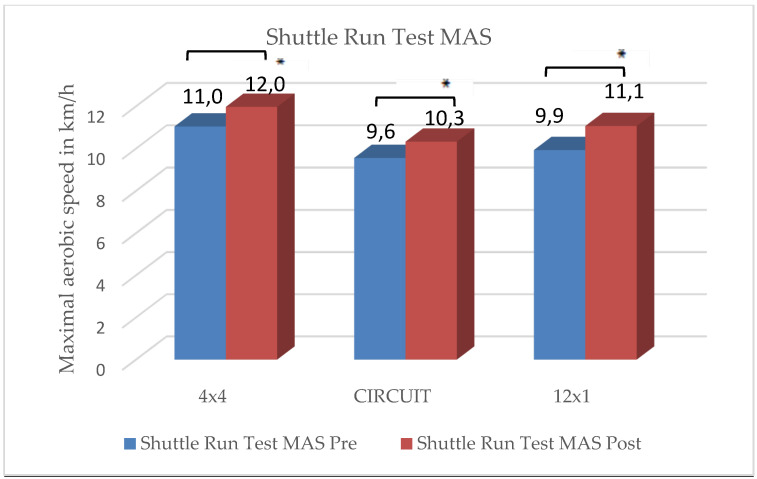
Changes in aerobic endurance capacity within intervention groups. * Significant difference between pre- and posttest (*p* < 0.05).

**Figure 3 ijerph-19-06855-f003:**
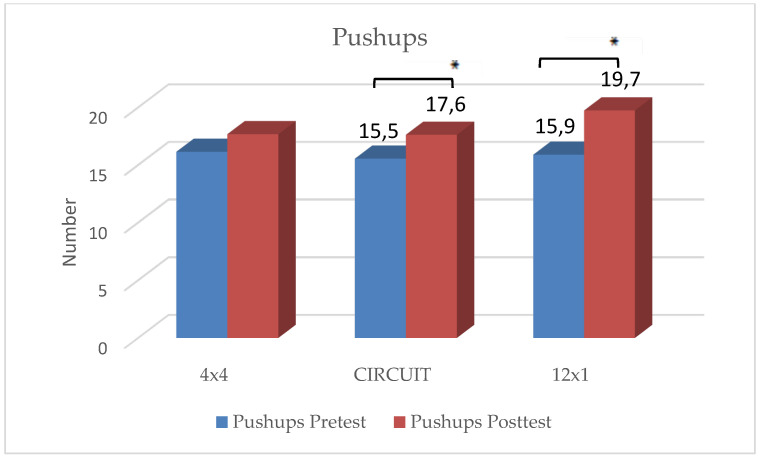
Changes in number of pushups within intervention groups. * Significant difference between pre- and posttest (*p* < 0.05).

**Figure 4 ijerph-19-06855-f004:**
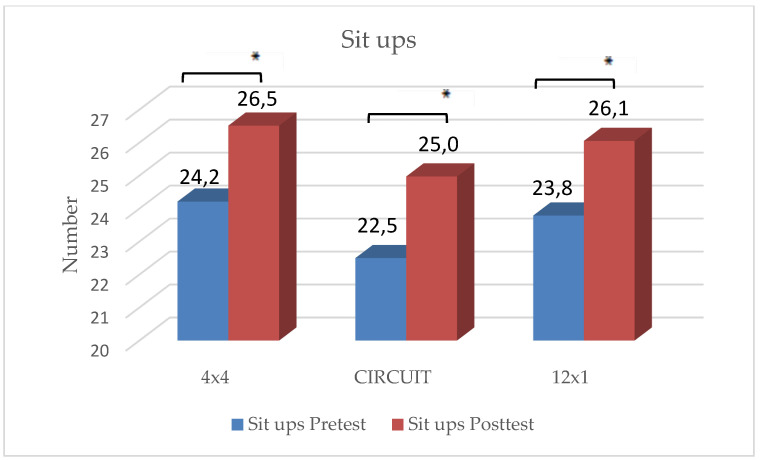
Changes in number of sit-ups within intervention groups. * Significant difference between pre- and posttest (*p* < 0.05).

**Figure 5 ijerph-19-06855-f005:**
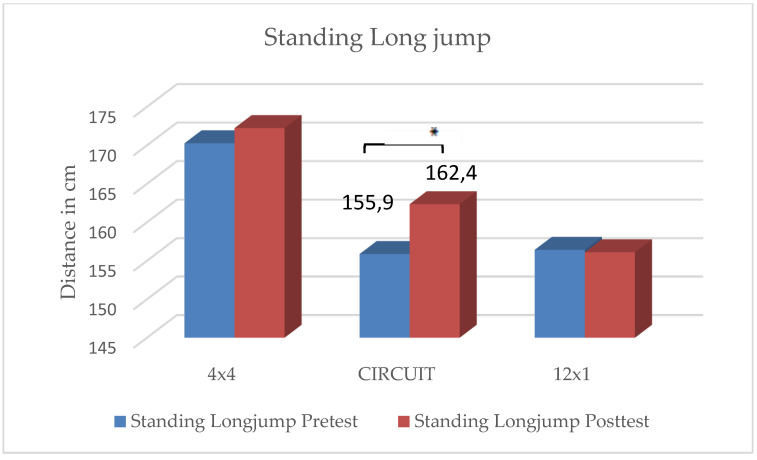
Changes in distance covered with standing long jump within intervention groups. * Significant difference between pre- and posttest (*p* < 0.05).

**Figure 6 ijerph-19-06855-f006:**
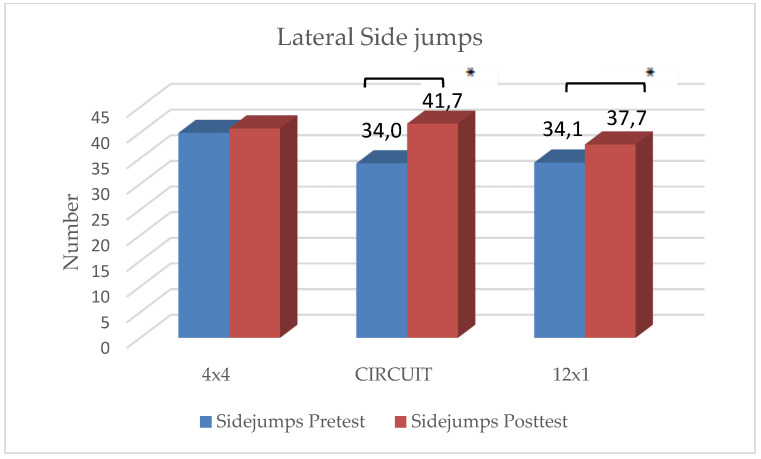
Changes in lateral side jumps within intervention groups. * Significant difference between pre- and posttest (*p* < 0.05).

**Table 1 ijerph-19-06855-t001:** Mean values and standard deviation of performance parameters evaluated via pre- and posttests for all three intervention groups.

	**4 × 4 HIIT (*n* = 41)**		**CIRCUIT (*n* = 40)**		12 × 1 HIIT (*n* = 40)		ANOVA		ANOVA		ANOVA	
							(Time of Measurement)	Power	(Group)	Power	(Time of Measurement × Group)	Power
Sex	23 Boys	17 Girls	22 Boys	18 Girls	24 Boys	17 Girls						
Mean age	13.3		11.9		12.2							
Parameter	Pretest	Posttest	Pretest	Posttest	Pretest	Posttest	df;F	ŋ^2^	df;F	ŋ^2^	df;F	ŋ^2^
Endurance Capacity (MAS)	11.05 +/− 2.24	11.97 * +/− 2.19	9.56 +/− 1.48	10.33 * +/− 1.68	9.93 +/− 1.51	11.07 * +/− 1.52	1; 123.99	0.549	2; 8.03	0.136	2; 1.37	0.026
Lateral side jumps	41.2 +/− 5.28	40.9 +/− 5.97	33.83 +/− 5.25	40.86 * +/− 7.28	37.44 +/− 6.50	42.33 +/− 6.86	1; 36.82	0.379	2; 2.09	0.087	2; 11.42	0.342
Pushups	16.15 +/− 7.20	17.67 +/− 8.53	15.55 +/− 6.27	17.63 * +/− 6.53	15.89 +/− 4.74	19.74 * +/− 6.59	1; 19.91	0.181	2; 0.24	0.005	2; 1.29	0.028
Sit-ups	24.21 +/− 6.08	26.52 * +/− 7.63	22.50 +/− 6.27	24.97 * +/− 6.87	23.79 +/− 5.91	26.05 * +/− 5.70	1; 14.47	0.143	2; 0.71	0.016	2; 0.012	0.000
Standing Long jump	170.26 +/− 23.50	172,24 +/− 28.90	155.88 +/− 20.53	162.35 * +/− 20.91	156.42 +/− 11.58	156.12 +/− 15.46	1; 3.62	0.039	2; 4.34	0.088	2:2.08	0.044

* Significant difference between pre- and posttest (*p* < 0.05); dF = degrees of freedom; ŋ^2^ = partial eta squared.

## Data Availability

The data presented in this study are available on request from the corresponding author.

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
