# Peer review of "Effects of Different Types of High-Intensity Interval Training (HIIT) on Endurance and Strength Parameters in Children and Adolescents"

_ijerph, 2022, doi:10.3390/ijerph19116855_

Round 1
Reviewer 1 Report
This work is devoted to a rather interesting issue - the possibility of conducting high-intensity interval training in school lessons and evaluating various options for such training. At the same time, both indicators of endurance and strength were studied.
However, the presented manuscript contains significant methodological limitations that should be eliminated.
Major:
- The Methods section indicates that the six participating classes were randomly assigned into one of the three training groups. At the same time, the Restrictions section states that it was impossible to randomly assign students according to their initially documented aerobic fitness levels. It is necessary to clarify what the randomization procedure was then when students were included in the study.
- Based on the design, students of different ages were included in the study (possibly, and with different gender, the latter is not indicated in the text of the manuscript). Since the ability of students to improve aerobic endurance and strength tests varies among people of different sex and age [Baquet G, et al], it is necessary to indicate how the groups differed or did not differ in these parameters. In the case of significant differences in these indicators, more sophisticated methods of statistical evaluation of the study results will be required (for example, multiple linear regression to assess the dynamics of each of the tests).
- The authors note that some of the students did not fully complete the program of training and tests. It is necessary to indicate in which groups they were originally assigned. It is also necessary to indicate the number of students in each group, both initially and those who completed the study. It is best to present this in the form of an appropriate flowchart.
- The authors also use a very vague term for student self-assessment - "athletic" and "unathletic". First, I would like to define more precisely the characteristics of these groups of students (in the Methods section). Secondly, since this characteristic of students influenced the dynamics of tests after a course of training, it is necessary to present the distribution of “athletic” and “unathletic” students in different groups.
- The list of references contains many references to old studies and few works from the last 5 years. In any case, it seems to me that it is necessary in the article in the discussion section to consider the recently published meta-analysis (Bauer N, et al).
Minor:
- Since this article is scientific and not popular, then table 3 should be in the Results section, and not in the appendix.
- Accordingly, tables 1-2 can be moved to the appendix.
- Figure 1 with the design of the study needs to be revised to be included in the flowchart (see above).
- If table 3 is finalized, then part of the information from the text should be moved there, it is also possible to consider the feasibility of having figures 2-5, whether there is a need to duplicate information in the table and in the figures.
References:
- Baquet G, Berthoin S, Gerbeaux M, Van Praagh E. High-intensity aerobic training during a 10 week one-hour physical education cycle: effects on physical fitness of adolescents aged 11 to 16. Int J Sports Med. 2001 May;22(4):295-300. doi: 10.1055/s-2001-14343.
- Bauer N, Sperlich B, Holmberg HC, Engel FA. Effects of High-Intensity Interval Training in School on the Physical Performance and Health of Children and Adolescents: A Systematic Review with Meta-Analysis. Sports Med Open. 2022 Apr 11;8(1):50. doi: 10.1186/s40798-022-00437-8.
Author Response
First of all, thank you very much for the helpful feedback and comments.
- The six participating classes had been randomly assigned to the three training groups. The three teachers who taught the training sessions had drawn lots. However, due to the fact that the study took place in a field setting in regular P.E. lessons, it was not possible to assign each and every student into a training group with students who had the exact same aerobic fitness level. Therefore, as in previous studies that I added in the manuscript, slight differences in initial performance level had to be accepted.
- Information regarding gender and age has been added in the manuscript. Also, similar relevant studies have been added with regard to slightly different initial fitness levels of the groups.
- Additional information on dropouts have been added.
- Admittingly, the term “athletic” is rather vague, but has been included in the feedback form the students gave. The information has been added in the method part. The distribution of ‘athletic’ and ‘non-athletic’ students was similar between the three groups. There was no significant difference between the self-evaluation of the students in this regard.
- The suggested additional literature has been added.
- Table 3 has been moved to the main part.
- Tables 1-2 have been moved to the appendix.
- Figure 1 has been revised

Reviewer 2 Report
This is an original article about an intervention study on a non-running based High Intensity Interval Training in children and adolescent. The manuscript is well written and well structured. It deals with an interesting topic, where further studies are needed. I suggest to the authors small adjustments to enhance clarity and comprehension of the results.
- Please to add a table with general characteristics of the study sample (e.g. mean age, weight etc)
- For the presentation of the results (text, tables and graphs) of the intervention methods used follow the same order presented in table 1 on page 4
- Regarding the pre- and post-testes at page 5 please indicate the tests relating to strength capacity for more complete information.
- The tables reported in the appendix are important for the results: please move them to the text.
- In the appendix A: Training program delete the row 7 (Robe skipping): it is a repetition
- Page 6 lines 241-243 and page7 lines 255-258: move the data on t test in table 3
- Figures 2-5 please add a related legend to the significance test.
- There is no mention of any differences between boys and girls: a mention on this topic could be added in the discussion / conclusions section
- Please add a list of abbreviations
Author Response
First of all, thank you very much for the helpful feedback and comments.
- Additional information on age has been added. However, participating students had not been weighed.
- The order of the presented results has been altered as suggested.
- Tables from the appendix have been moved to the text
- The rope skipping exercise has been done several times in the circuit training group. The circuit consisted of twelve whole-body-drill, rope skipping was used several times.
- Legends have been added.

Round 2
Reviewer 1 Report
The authors of the manuscript did a great job of improving it, my comments were taken into account and the text was corrected. I have no more comments.